# Improved Colony Predation Algorithm Optimized Convolutional Neural Networks for Electrocardiogram Signal Classification

**DOI:** 10.3390/biomimetics8030268

**Published:** 2023-06-21

**Authors:** Xinxin He, Weifeng Shan, Ruilei Zhang, Ali Asghar Heidari, Huiling Chen, Yudong Zhang

**Affiliations:** 1School of Emergency Management, Institute of Disaster Prevention, Sanhe 065201, China; hexinxin0305@gmail.com (X.H.); zhangrl420@163.com (R.Z.); 2School of Surveying and Geospatial Engineering, College of Engineering, University of Tehran, Tehran 1417935840, Iran; as_heidari@ut.ac.ir; 3Institute of Big Data and Information Technology, Wenzhou University, Wenzhou 325000, China; 4School of Computing and Mathematical Sciences, University of Leicester, Leicester LE1 7RH, UK

**Keywords:** colony predation algorithm, convolutional neural networks, ECG, hyperparameter optimization, orthogonal learning strategy, swarm intelligence algorithm

## Abstract

Recently, swarm intelligence algorithms have received much attention because of their flexibility for solving complex problems in the real world. Recently, a new algorithm called the colony predation algorithm (CPA) has been proposed, taking inspiration from the predatory habits of groups in nature. However, CPA suffers from poor exploratory ability and cannot always escape solutions known as local optima. Therefore, to improve the global search capability of CPA, an improved variant (OLCPA) incorporating an orthogonal learning strategy is proposed in this paper. Then, considering the fact that the swarm intelligence algorithm can go beyond the local optimum and find the global optimum solution, a novel OLCPA-CNN model is proposed, which uses the OLCPA algorithm to tune the parameters of the convolutional neural network. To verify the performance of OLCPA, comparison experiments are designed to compare with other traditional metaheuristics and advanced algorithms on IEEE CEC 2017 benchmark functions. The experimental results show that OLCPA ranks first in performance compared to the other algorithms. Additionally, the OLCPA-CNN model achieves high accuracy rates of 97.7% and 97.8% in classifying the MIT-BIH Arrhythmia and European ST-T datasets.

## 1. Introduction

In recent years, research has shown that deep learning offers numerous benefits compared to conventional machine learning approaches [1,2]. Deep learning is known to extract features efficiently compared to traditional machine learning approaches, so many researchers tend to work with deep learning [3,4]. These methods are robust against noise, can achieve superior accuracy, and can be scaled straightforwardly to larger datasets, consequently reducing training time [5,6,7]. The main methods commonly used for deep learning are deep neural networks and generative adversarial networks. Among these networks, convolutional neural networks (CNNs) have made many contributions to the field of computer vision and are popular among researchers [8]. LeNet-5 [9] is the pioneer CNN, a convolutional neural network algorithm proposed by Yann LeCun in 1998 to solve the problem of handwriting recognition. Then, the AlexNet [10] network structure was introduced, and it won the 2012 ImageNet competition. Since then, CNNs have received widespread and enthusiastic attention worldwide, and more new network structures have been proposed, such as VGG [11], GoogleLeNet [12], ResNet [13], and DenseNet [14]. When these new network structures were created, the number of layers and parameters increased accordingly. However, tuning hyperparameters is a highly challenging task. Since the number of parameters is very large and specialized personnel are required to select the optimal parameters based on experience, this can result in the loss of a large volume of labor resources and the waste of material resources. Therefore, it is a tough task to manually tune parameters using manual labor with limited resources.

Optimization techniques range from multiobjective methods to single objective approaches to many objective techniques [15,16,17]. Each of these approaches has its own unique set of computational difficulties, making the process of optimization both challenging and rewarding [18,19]. The challenges of the optimization methods presented in the previous literature include a high computational cost, difficulty in tackling multiple local optima, lack of robustness, immature convergence, and conditionality of global optima results [20,21,22]. As one of the main classes of optimization methods, activity patterns of various groups of organisms are used to generate swarm intelligence (SI) algorithms [23]. In recent years, SIs have been able solve complex optimization problems in the real world due to their excellent optimization capabilities [24,25]. Some of the famous algorithms are the particle swarm algorithm (PSO) [26], Harris hawk optimization algorithm (HHO) [27], slime mould algorithm (SMA) [28,29], hunger games search (HGS) [30], Runge Kutta optimizer (RUN) [31], the weighted mean of vectors (INFO) [32], colony predation algorithm (CPA) [33], and rime optimization algorithm (RIME) [34]. They have been applied to solve many problems, such as bankruptcy prediction [35], economic emission dispatch [36], feature selection [37,38], constrained multiobjective optimization [39], dynamic multiobjective optimization [40], global optimization [41], medical image segmentation [42], feed-forward neural networks [43], scheduling optimization [44], large-scale complex optimization [45], multiobjective optimization [46], and numerical optimization [47]. Among them, CPA is a new algorithm proposed in recent years that is based on prey predation by animal groups in nature. CPA has a more vital optimization ability than PSO, MFO, and other algorithms [33]. Because swarm intelligence algorithms can solve complex practical problems, many researchers have proposed optimizing the parameters of deep learning network structures using SI. Researchers also need to assess the performance of swarm intelligence algorithms. IEEE CEC2017 serves as a benchmark function for testing the performance of such algorithms and comprises four categories: unimodal functions, multimodal functions, hybrid functions, and composition functions.

There are currently two types of researchers studying deep learning network structures. Some researchers use manually configured deep learning network structures, while others use hyperparameters that use SIs to optimize deep learning network structures. Pyakillya et al. [48] proposed a model for automatic classifications using a deep learning architecture that consists of a one-dimensional convolutional layer and a fully connected layer. The model achieved an accuracy of 86% on the validation dataset. Mathews et al. [49] proposed a deep learning model for electrocardiogram (ECG) classification that incorporates a restricted Boltzmann machine and a deep belief network. The experiments showed that this model performed better at low sampling rates. Sannino et al. [50] proposed a deep neural network with seven hidden layers for automatic classification and verified experimentally that this model outperforms other models in terms of accuracy, achieving a precision of 99.52%. Strodthoff et al. [51] proposed a deep learning-based time series classification algorithm that mainly uses the ResNet network model, and the experimental results proved that the performance of this algorithm is promising. Peimankar et al. [52] proposed a method for ECG signal detection composed of deep learning-based convolutional neural networks and long- and short-term memory. Different heartbeat waveforms were detected from the MITDB and QTDB datasets to test the method’s effectiveness. The F1 scores obtained on the two datasets are 99.56% and 96.78%, respectively, indicating that this method was very effective for ECG signal detection. Hasan et al. [53] proposed the use of one-dimensional convolutional neural networks for the recognition of multiple heart diseases, and the accuracy of this method on the MIT-BIH, St-Petersberg, and PTB datasets is 97.7%, 99.71%, and 98.24%, respectively. Acharya et al. [54] investigated a nine-layer convolutional neural network structure for identifying five different classes of heartbeats in ECG signals, and the experimental results showed that the accuracy was 94.03%.

Compared with manual search, automatic search using SI can be performed by improving the algorithm to find the values of suitable parameters within the selected range and finally find the most optimal values.

In the study of Houssein et al. [55], a convolutional neural network model based on an improved marine predators algorithm (IMPA-CNN) was proposed to exploit the best classification role of CNNs and to find the best hyperparameters of CNNs. This model uses the improved marine predators algorithm to select the most suitable CNN parameters automatically, and the experimental results of testing it on different ECG datasets show that this model is very effective. Khalifa et al. [56] proposed a method to optimize the parameters of a seven-layer CNN network; the first six layers of the CNN use gradient descent and the last layer uses a particle swarm algorithm to find the optimal parameters of the CNN. This model was compared with a handwritten character recognition dataset using a standard CNN architecture, and the results show that this model has a higher accuracy, with a value of 96.67%. Yamasaki et al. [57] proposed a method to improve image recognition accuracy, in which a particle swarm algorithm is implemented to automatically find the appropriate hyperparameters of the CNN. The results show higher accuracy when using this method to optimize the AlexNet structure and compare it with the standard AlexNet-CNN structure in image recognition experiments. Dey et al. [58] proposed a model integrating three network structures, VGG19, ResNet50, and DenseNet121, used to screen tuberculosis or TB images from chest X-rays. To overcome the problem of manual tuning, the training part of the model uses an optimization algorithm to set the parameters of the model, and this method was proven effective for TB classification by testing on a TB dataset.

Using convolutional neural networks, Pathan et al. [59] investigated a method to automatically identify chest X-ray images affected by COVID-19. This method achieved 98.8% and 96% accuracy for dataset 1 and dataset 2, respectively. The results of the experiments on COVID-19 images show that this method can effectively screen out patients with the disease. The hyperparameters of the DenseNet121 architecture were optimized using the gravity search algorithm in the work of Ezzat et al. [60]. This optimized model was compared with the CNN architecture of Inception-v3 in experiments for the detection of new crown pneumonia, and the experimental results indicate that this model performs exceptionally well in terms of accuracy, reaching 98.38%, significantly higher than its competitor. Most studies used optimization algorithms to tune the parameters of CNNs, while some works used optimization algorithms to tune the overall architecture of CNNs to select the most appropriate number of network layers. In Singh et al. [61]′s study, a multi-stage particle swarm was used to optimize the network structure and hyperparameters of CNNs, which is divided into two stages. In the first stage, the multi-level particle swarm algorithm is used to optimize the CNN architecture and determine the number of network layers that can better exploit the performance of the CNN. In the second stage, the hyperparameters are tuned based on this network structure. The final model was tested using an image dataset, and the results show that the performance of this model is excellent. To speed up finding the layers and parameters of CNN architecture, Fernandes et al. [62] proposed a new particle swarm velocity operator and used this new particle swarm algorithm to optimize the architecture and parameters of a CNN. Experimental tests show that this model can find an optimized CNN model based on any dataset.

Although many researchers have studied this area, there are still many challenges to be tackled. CPA faces the same challenges as other swarm intelligence algorithms, such as falling into local optima and slow convergence. To solve these problems, CPA needs to be improved. Therefore, this motivates us to propose an improved CPA and use it to optimize the parameters of a CNN.

This paper proposes an improved CPA based on an orthogonal learning strategy (OLCPA). To demonstrate the effectiveness of OLCPA in optimizing CNN parameters, it was applied to the classification of arrhythmia in ECG datasets. The main contributions of this paper are as follows:An OLCPA algorithm based on the orthogonal learning strategy is proposed, and it is compared with four traditional and seven advanced algorithms on the IEEE CEC 2017 benchmark functions.This paper analyzes the scalability of OLCPA and CPA on different dimensions of the IEEE CEC2017 benchmark functions.A CNN-based OLCPA-CNN model for identifying abnormal ECG signals is designed.The OLCPA-CNN model is compared with other methods using the MIT-BIH and the European ST-T datasets.

The rest of this paper is as follows: Section 2 describes CPA and introduces the background knowledge on CNNs. Section 3 describes the improved CPA algorithm (OLCPA). The process of the OLCPA-CNN model is described in Section 4. Section 5 describes the experimental design and results of OLCPA. Section 6 describes the application of OLCPA-CNN. Section 7 is the discussion. Finally, Section 8 is the conclusion of this paper.

## 2. Preliminary Work

### 2.1. Overview of Colony Predation Algorithm

The colony predation algorithm [33] was inspired by the fact that cooperative communicative group predation of group-housed animals increases the probability of successful predation.

Group-living animals pursue their prey by communicating with each other, and Equation (1) simulates this process.
(1)$$X_{j}^{i}\left(t+1\right)=X_{j}^{i}\left(t\right)+\left(1-r\right)\cdot \left(\frac{X_{1}\left(t\right)+X_{2}\left(t\right)}{2}\right)$$
where $X_{j}^{i}\left(t\right)$ is the individual currently searching for prey in the j-th dimension, j ranges from 1 to dim, and *i* represents the current individual. $X_{1}$ and $X_{2}$ are the positions of the two individuals closest to the prey, r is a random number between 0 and 1, and $X_{j}^{i}\left(t+1\right)$ is the position of the individual in the next iteration.

Two strategies are used in the pursuit process to increase the probability of successful predation: scattering prey and surrounding prey. Prey dispersal is to drive the prey in different directions and weaken the prey group, and Equation (2) simulates this process.
(2)$$X\left(t+1\right)=X_{best}-S\cdot \left(r_{1}\cdot \left(ub-lb\right)+lb\right)$$
where *S* denotes the energy of the prey and its value is changed, *lb* and *ub* are the left and right values of the boundary range, $r_{1}$ is a random number between 0 and 1, $X_{best}$ is the location of the prey, and $X\left(t+1\right)$ is the current position of the pursuer. The variation of *S* is as follows:(3)$$S_{0}=a-t\cdot \left(\frac{a}{N}\right)$$
(4)$$S=2\cdot S_{0}\cdot r_{2}$$
(5)$$a=e^{w-2w\left(1-\frac{t}{MaxFes}\right)}$$
where $r_{2}$ varies between 0 and 1, *t* is the current number of evaluations, and *N* indicates the number of predators. $S_{0}$ varies with the number of evaluations, the value of *a* is related to the number of evaluations, and the value of *w* is 9.

After the prey has been successfully dispersed, the predators use a siege attack on them, a process shown in Equation (6):(6)$$X\left(t+1\right)=X_{best}-2S\cdot D\cdot e^{l}\cdot \tan\left(\frac{\pi }{4}l\right)$$
(7)$$D=|X_{best}-X_{\left(t\right)}|$$
where *D* is the interval indicating the current individual’s distance from the prey. The probability of the two strategies being executed is shown in Equation (8).
(8)$$X\left(t+1\right)=\left\{\begin{array}{c} X_{best}-S\cdot \left(r_{1}\cdot \left(ub-lb\right)+lb\right)\;r\geq 0.5\\ X_{best}-2S\cdot D\cdot e^{l}\cdot \tan\left(\frac{\pi }{4}l\right)\;r<0.5 \end{array}\right.$$

When the predator begins to chase prey, there are two strategies: one is when the predators find prey nearby, and choose to support the closest predator to the prey; Equation (9) simulates this process. The second is when the predators do not find prey around them, they will randomly choose other locations of prey; this process is shown in Equation (11).
(9)$$X\left(t+1\right)=P_{nearest}$$
(10)$$D1=\left|2r_{4}\cdot X_{rand}-X\left(t\right)\right|$$
(11)$$X\left(t+1\right)=X_{rand}-S\cdot D_{1}$$
(12)$$X_{rand}=r_{5}\cdot \left(\left(ub-lb\right)+lb\right)$$
where $P_{nearest}$ denotes the position of the individual closest to the prey, $D_{1}$ denotes the distance moved by the random population, and $X_{rand}$ denotes the new position randomly generated by the prey individuals.

The probability of the above two strategies being executed is described in Equation (13). Algorithm 1 describes the process of implementing CPA.
(13)$$X\left(t+1\right)=\left\{\begin{array}{c} P_{nearest}\;|r_{6}|\leq 1\\ X_{rand}-S\cdot D_{1}\;|r_{6}|>1 \end{array}\right.$$

**Algorithm 1** Pseudo-code for CPAInitialize population size *Num*, the problem dimension *dim*, and the maximum number of evaluations *MaxFes*
**While** (*t* ≤ *MaxFes*)**For** *i* = 1: *Num*
Calculation of individual fitness valuesUpdate Xbest**End for**   **For** *j* = 1: *dim*   Update X1, X2   Calculate Xji using Equation (1)   **End for**   **For** *i* = 1: *Num*   Update *S*   **If** |S|<23a   Calculate the current agent’s position by Equation (8)      **End if**      **If** |S|≥23a   Calculate the current agent’s position by Equation (13)      **End if**   **End for**
   *t = t* + 1**End while**Return Xbest


### 2.2. Convolutional Neural Network

Yann Lecun of New York University proposed the convolutional neural network in 1998. It differs from a multilayer perceptron (MLP) and is used in various fields, such as image processing. The difference with regard to MLP is that the way of CNN local connection and weight sharing is changed, where on the one hand, the network can be better optimized by reducing the number of weights, and on the other hand, the complexity of the model can be effectively reduced. The convolutional neural network structure is a deep neural network with a convolutional structure, and its overall architecture of network structure includes an input layer, convolutional layer, rectified linear units (ReLU) layer, pooling layer, and fully connected layer. In practical applications, the convolutional layer contains the convolutional layer and the ReLU layer. Activation functions are usually used to compute the convolutional layer, and the commonly used activation functions are the Sigmoid, Tanh, and ReLU functions. The pooling layer is generally arranged after the convolutional layer to reduce the network’s parameters and computational resources. The role of the fully connected layer in the entire network is to classify, and it is usually found in the last few layers of the CNN. The one-dimensional CNN structure of this paper is shown in Figure 1.

## 3. The Improved CPA

CPA is an algorithm with better optimization performance. However, when faced with complex optimization problems, it tends to fall into local optima or slow down the convergence rate due to the fact that it lacks some strategies that can flexibly address these problems. To overcome these problems, we propose an improved CPA that incorporates an orthogonal learning strategy.

### 3.1. Orthogonal Learning Design

An orthogonal learning design [63] is a method that uses a small number of experiments to find the best solution. The determination of the small number of experiments is mainly related to two factors in the orthogonal table $L_{M}$: the *K* factor and the number of *Q* levels for each factor. $L_{M}$ ($Q^{K}$) indicates that $Q^{K}$ sets of experiments need to be carried out, but when the values of *Q* and *K* are large, it is impossible to complete this many experiments, so it is necessary to use the orthogonal table to design the number of experiments. For experiments designed using the orthogonal table, only *M* combinations need to be chosen to complete the experiments, and the number of experiments *M* is much smaller than $Q^{K}$. For example, the following orthogonal table $L_{9}$($3^{3}$) explains the process.
(14)$$L_{9}(3^{3})=\left[\begin{array}{ccc} 1 & 1 & 1\\ 1 & 2 & 2\\ 1 & 3 & 3\\ 2 & 1 & 2\\ 2 & 2 & 3\\ 2 & 3 & 1\\ 3 & 1 & 3\\ 3 & 2 & 1\\ 3 & 3 & 2 \end{array}\right]$$

In Equation (14), $L_{9}$ indicates that the experiment with the orthogonal table design needs to be executed only 9 times, but without the orthogonal design, this experiment needs to be executed 27 times. Therefore, the orthogonal design can significantly reduce the number of experiments, and this method is more effective when the values of *Q* and *K* are larger.

### 3.2. Orthogonal Learning Strategy

This study introduces the orthogonal learning strategy (OL) into CPA, which uses orthogonal tables to generate a new search mechanism to explore more regions and avoid becoming trapped in local optima. After using this strategy, the original CPA generates M + 1 search agents, which can improve the exploration ability of CPA.

### 3.3. The Proposed OLCPA

This subsection proposes a novel CPA algorithm based on an orthogonal learning strategy. Equation (15) describes the formation process of OLCPA. This new OLCPA algorithm exploits the orthogonal strategy’s search mechanism to expand the solution’s search scope and find high-quality solutions. The pseudo-code for OLCPA is described in Algorithm 2, and Figure 2 describes the specific process of OLCPA.
(15)Xt+1=XnewFXnew<F(Xold)Xoldothers
where Xnew represents the new search agent generated using the orthogonal policy and Xold is the search agent without the orthogonal policy. *F* represents the fitness function.
**Algorithm 2** Pseudo-code for OLCPAInitialize population size *Num*, the problem dimension *dim*, and the maximum number of evaluations *MaxFes***While** (*t* ≤ *MaxFes*)**For** *i* = 1: *Num*Calculation of individual fitness valuesUpdate Xbest**End for**   **For**
*j* = 1: *dim*   Update X1, X2   Calculate Xji by Equation (1)   **End for**   **For** *i* = 1: *Num*   Update *S*   **If** |S|<23a   Calculate the current agent’s position by Equation (8)      **End if**      **If** |S|≥23a   Calculate the current agent’s position by Equation (13)      **End if**   **End for**
   **Execute an orthogonal strategy**   **Update the current search agent**   *t = t + 1***End while**Return Xbest


## 4. The Design of the OLCPA-CNN Model

This section describes the OLCPA formation process, and the classification error rate of CNN is used as the fitness function. Then, the parameters of the CNN are optimized according to the optimal solution generated by OLCPA to obtain the OLCPA-CNN model and evaluate this model.

### 4.1. The Network Structure of CNN

The nine-layer CNN model used in this paper is displayed in Figure 1, and the overall structure consists of three sets of convolutional pooling layers and two fully connected layers, where each set of convolutional pooling layers consists of one convolutional layer and one pooling layer. Data are transmitted to the input layer, which undergoes convolution and pooling operations to achieve the mapping of different functions and simultaneously extract useful features, and then achieves the classification purpose in the fully connected layer. Figure 3 depicts the process of OLCPA-CNN being designed. The search agent of OLCPA represents the parameters of the CNN, and the optimized parameters of the CNN are obtained through iterative updates of the position of the search agent of OLCPA.

### 4.2. Hyperparameter Optimization

Parameter tuning occupies a crucial place in deep learning classification [64,65]. OLCPA optimizes the parameters of the CNN structure by iteratively updating the solution. In this paper, the parameters to be optimized are concentrated in the following layers: a three-layer convolutional layer and a two-layer fully connected layer. The optimization of hyperparameters in the CNN includes the number and size of convolutional kernels for each convolutional layer, the number of first fully connected layers, the learning rate, the number of epochs, and L2 regularization. The optimized parameters and their ranges are described in Table 1. The parameters of OLCPA are as follows: the number of populations is 5, the maximum number of iterations is 20, and the dimensionality is set to 10. Because the number of parameters to be optimized is closely related to the parameters of the CNN, its bounded range is related to the range of parameter variations.

### 4.3. Fitness Function

The fitness function is essentially an evaluation function that evaluates the goodness of an algorithmic solution by measuring the fitness of individuals in the population [66]. In this article, the fitness function is the classification error rate, closely related to classification accuracy (ACC). The correct classification rate is calculated based on the confusion matrix. Table 2 shows the expressions of the confusion matrix, where TP means that both the actual and predicted values are positive, TN means that both values are negative, FN means that the predicted value is negative and the real value is positive, and FP means that the predicted value is positive and the real value is negative. The formula for ACC is shown in Equation (16), and the corresponding fitness function is calculated as shown in Equation (17). The relationship between the value of the fitness function and the classification error rate is proportional; when the value of the fitness function becomes smaller, the classification error rate also becomes smaller, which reflects that the quality of the individual solution is good and it is beneficial to the optimization of CNN parameters.
(16)$$ACC=\frac{TP+TN}{TP+FP+FN+TN}$$
(17)Fitness=1−ACC

## 5. Experimental Design and Results of OLCPA

In this section, we compare the OLCPA algorithm with other algorithms on the IEEE CEC2017 benchmark functions to verify the performance of OLCPA. These algorithms were tested using MATLAB R2018b with 128GB of RAM and an Intel(R) Xeon(R) Silver 4110 CPU on Windows Server 2016. The assessment of AI-based approaches through fair procedures can advance replicability, transparency, research standards, and public confidence [67,68,69]. Comparing computational methods using the same criteria allows us to established unbiased assessments [70,71]. We conducted our trials in a manner consistent with fair comparison principles. The parameters involved in the experiments are as follows: the number of populations N, dimension D, the maximum number of evaluations MaxFes, and the number of independent runs of the algorithm Num. Table 3 shows the values of these parameters.

### 5.1. Benchmark Function

In this subsection, the IEEE CEC2017 benchmark functions [72] are used to test the performance of the OLCPA. These functions are classified into the following categories: unimodal functions (F1–F3), multimodal functions (F4–F10), hybrid functions (F11–F20), and complex functions (F21–F30).

### 5.2. Scalability Test

This section tests OLCPA and CPA in 50 and 100 dimensions under the same experimental conditions. The dimensionality tests for scalability are mainly to verify the performance of OLCPA when coping with different dimensions. Table 4 shows the comparison results of these two algorithms in different dimensions. Avg and Std denote the mean and standard deviation of the experimental results, respectively. The experimental results reveal that OLCPA can also perform well in high dimensionality. The experimental results in the table show that the quality of most solutions of OLCPA is significantly stronger than that of CPA in 50 and 100 dimensions, which indicates that the improved OLCPA algorithm based on CPA is effective and the performance of OLCPA is stronger.

### 5.3. Comparison with Conventional and Advanced Algorithms

In this section, to test the performance of OLCPA, it is compared with 11 algorithms: CCMWOA [73], IGWO [74], CCMSCSA [75], BMWOA [76], CMFO [77], CESCA [78], GCHHO [79], DE [80], MFO [81], HGS [30], and CPA [33]. To ensure the reliability of the experiment, this experiment was conducted under the same experimental conditions. The results of comparing OLCPA with the algorithms mentioned above are listed in Table 5. The last three columns of the table are Rank, the symbol “+/=/−”, and Avg, where Rank represents the Friedman test, “+/=/−” represents the number of functions in which OLCPA is stronger than, equal to, or not stronger than the other algorithms for the 30 benchmark functions, and Avg represents the average of the benchmark function test results.

From the experimental data in Table 5, we can see that OLCPA ranks first and has a smaller mean value of 3.1322 compared to the mean value of 3.6933 for CPA, which reflects the stronger effect of OLCPA than CPA. It can be seen that OLCPA outperformed CCMWOA and CESCA for all 30 functions on CEC 2017. Although the performance of CMSCSA is close to that of OLCPA, OLCPA performs better for multimodal and mixed modes, and OLCPA is the first among these competitors.

From the results of the Wilcoxon signed-rank test in Table 6, the *p*-values of most algorithms are less than 0.05, proving the statistically superior performance of OLCPA compared to other algorithms.

Figure 4 depicts the convergence curves of OLCPA and the other algorithms, and it can be seen from these convergence function plots that OLCPA has the best ability to find the optimal solution compared to the other algorithms. Although the competition between HGS and OLCPA is fierce in functions F4, F12, and F19, OLCPA converges with increasing speed and accuracy in the later iterations and finally finds the optimal solution. Based on the trend of these convergence plots, it can be seen that the convergence speed and accuracy of OLCPA are better than those of the competitors.

## 6. Application in ECG Signal Classification

### 6.1. Test Datasets

This section focuses on the datasets used for training and testing. PhysioNet is a well-known research resource for studying complex physiological signals, including the MIT-BIH Arrhythmia Database [82] and the European ST-T Database [83], used for testing and training experiments.

#### 6.1.1. MIT-BIH Arrhythmia Database

Data in the MIT-BIH database were obtained from the ECG recordings of 47 patients, 60% of whom were inpatients and 40% of whom were outpatients, tested by the Boeheim Arrhythmia Laboratory from 1975 to 1979. These records consist of 48 half-hour dual-channel ECG recordings; each record was sampled at a rate of 360 Hz, and the two channels were modified limb leads (MLII) with V channels from V1 to V5.

#### 6.1.2. European ST-T Database

The European ST-T database contains mainly ST and t-wave variations, and the data are derived from 90 ECG recordings from 79 test subjects, who were men between 30 and 84 years of age and women between 55 and 71. Each recording spanned 2 h and contained two signals, each with a sampling rate of 250 Hz.

Due to the need to select an appropriate sample size, the MIT-BIH database was classified into four categories: N, VEBs, Q, and S, following the AAMI standard classification approach. The AAMI classification approach is described in Table 7. Due to the uneven sample distribution, four categories (N, S, Q, VEB) from the AMMI classification were selected for the MIT-BIH database for testing, and three (N, S, VEB) were selected for the European ST-T database.

In our experiment, the number of European ST-T datasets used is 3000, of which 2000 were in the training set and 1000 were in the test set, and the number of MIT-BIT datasets used is 10,000, of which 8000 were in the training set and 2000 were in the test set. Table 8 shows the number of samples per category for the datasets.

### 6.2. Metrics for Performance Evaluation

The evaluation metrics used in this section differ from those used in Section 5. In Section 5, mean and standard deviation were utilized to evaluate the performance of the OLCPA algorithm, which is a continuous problem. However, this section mainly focuses on utilizing the OLCPA-CNN model to solve ECG electrocardiogram signal classification problems, which are categorical problems and differ from continuous problems. Therefore, in order to effectively evaluate the performance of the OLCPA-CNN model and compare it with methods used by other researchers, common metrics in deep learning such as accuracy (ACC), precision (*Pr*), specificity (*Sp*), sensitivity (*Se*), and F-score (F1) are utilized. They are calculated as follows:(18)$$Se=\frac{TP}{TP+FN}$$
(19)$$Sp=\frac{TN}{FP+TN}$$
(20)$$Pr=\frac{TP}{TP+FP}$$
(21)$$F1=\frac{2TP}{2TP+FP+FN}$$
where the meanings represented by *TP*, *TN*, *FP*, and *FN* are described in Section 5.3.

ACC indicates the rate of correct classification. The value of ACC can reflect the good or bad performance of the model classification; when the value of the former is large, accordingly, the performance of the latter is also good. The sensitivity indicates the number of positive samples as a percentage of the number of all true positive samples. The sensitivity value is proportional to the classification accuracy of positive samples, and when the value of the former increases, the latter also increases accordingly. Specificity indicates the number of samples predicted to be negative among all true negative samples. Precision indicates the proportion of true positive samples among all positive samples. The F1 integrally evaluates the performance of a classifier, and the larger the F1, the better the performance of this classifier.

### 6.3. Performance Analysis of OLCPA-CNN on Datasets

The proposed OLCPA-CNN model was tested and trained using two different datasets. Since OLCPA can tune the parameters of the CNN through the iterative update of the population, it allows the CNN to play a better role in classification. The classification accuracy of this model on the two datasets (MIT-BIH, ST-T) is 97.9% and 97.8%, respectively, which shows that the performance of this OLCPA-CNN model is excellent.

Figure 5 depicts the average values of the evaluation metrics for OLCPA-CNN, CPA-CNN, and a randomly generated CNN on the MIT-BIH dataset; the results in the figure indicate that the evaluation metrics of OLCPA-CNN are higher than those of CNN and CPA-CNN, which indicates that OLCPA-CNN outperforms CPA-CNN. Figure 6 shows the average performance of OLCPA-CNN, the randomly generated CNN, and CPA-CNN on the ST-T dataset. The results in the figure indicate that the performance of OLCPA-CNN is significantly stronger than that of CPA-CNN and the other CNN, which reflects that the overall optimization of OLCPA-CNN is good. Figure 7 and Figure 8 show the correct and loss rates of OLCPA-CNN on the ST-T and MIT-BIH datasets, respectively, and the trend of the curves shows that as the number of iterations increases, the correct rate approaches 100% and the loss rate converges to 0.

To highlight the effective performance of OLCPA-CNN on the MIT-BIH dataset for classification performance, OLCPA-CNN was compared with other methods. These methods include the following: Li et al. [84] proposed a model for optimizing support vector machine classifiers using a genetic algorithm and used it for the MIT-BIH dataset; Patro et al. [85] proposed optimizing machine learning using optimization algorithms and applied this to the MIT-BIH dataset, where the algorithms and classifiers involved include the support vector machine (SVM) and random forest (RF), genetic algorithm (GA) and particle swarm algorithm (PSO); Acharya et al. [29] investigated a nine-layer convolutional neural network structure for identifying five different classes of heartbeats in ECG signals. The experimental data in Table 9 reflect that OLCPA-CNN is effective compared with these other methods.

## 7. Discussion

The classification results demonstrate that OLCPA-CNN can automatically search for the best hyperparameters suitable for a CNN, and the proposed OLCPA-CNN model can effectively address ECG classification tasks. Moreover, the ability to automatically extract features and perform classification is precious, particularly considering the significant expense associated with manual feature annotation by specialized professionals. However, when optimizing complex network architectures and handling vast amounts of data, this technique also has the drawback of high time costs. In addition, there is currently no universally applicable solution to every problem, and it is necessary to decide on the optimization algorithm and network architecture—including hyperparameters, number of neurons, and layers—to use based on the specific problem being addressed. Therefore, to solve ECG classification problems, this study selected an improved CPA algorithm, namely the OLCPA optimization algorithm, to optimize the hyperparameters of a CNN. Of course, in future work, it can also be applied to more cases, such as the optimization of machine learning models [86], fine-grained alignment [87], computational experiments [88,89], Alzheimer’s disease identification [90], iris or retinal vessel segmentation [91,92], MRI reconstruction [93], service ecosystem [94], structured sparsity optimization [95], tensor recovery [96,97], medical image computing [98], computer-aided medical diagnosis [99], image denoising [100], renewable energy generation [101], and medical signals [102,103].

## 8. Conclusions and Future Works

This article presents an OLCPA algorithm based on orthogonal learning strategies and proposes an OLCPA-CNN model that optimizes the hyperparameters of a CNN using the OLCPA algorithm. The experimental results show that the OLCPA-CNN model achieves excellent classification performance, with an accuracy of 97.90% on the MIT-BIH dataset, outperforming other models proposed by researchers. As the MIT-BIH dataset is a type of time-series data, the OLCPA-CNN model presented in this paper can be used for ECG classification and other time-series datasets, such as geomagnetic data. However, the use of this model is limited and it may not be suitable for different classification problems. Therefore, specific algorithms should be designed, and appropriate network structures should be optimized for specific problems.

In future work, reducing the running time will be the focus due to the drawback of this parameter optimization process being a waste of time. Secondly, this model is also valuable for other problems, such as regression problems. Third, CPA can be combined with other network structures.

## Figures and Tables

**Figure 1 biomimetics-08-00268-f001:**
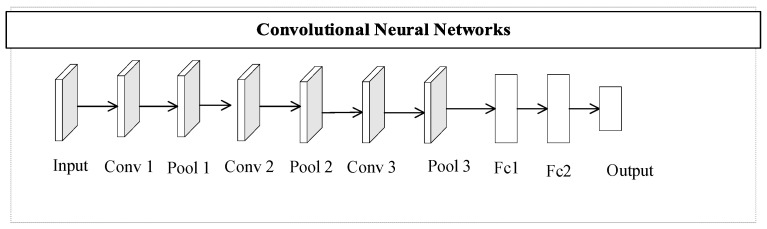
The architecture of a nine-layer CNN.

**Figure 2 biomimetics-08-00268-f002:**
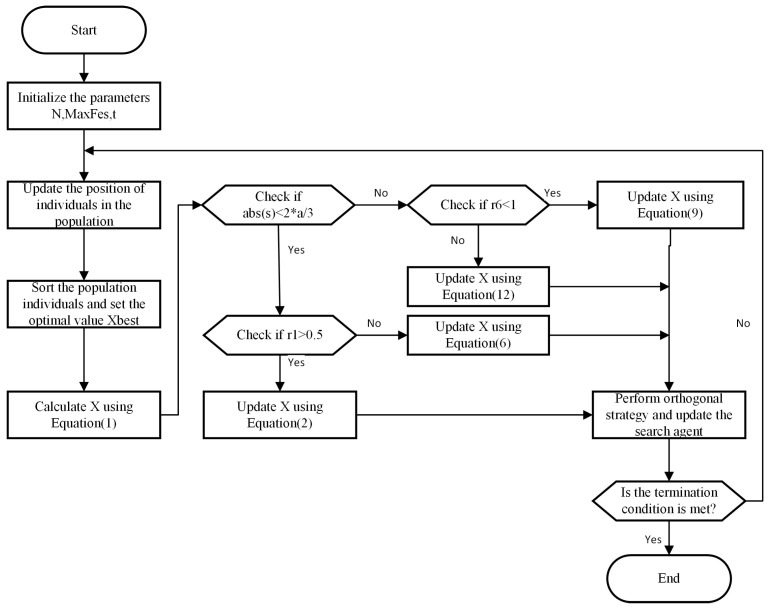
Flowchart of OLCPA.

**Figure 3 biomimetics-08-00268-f003:**
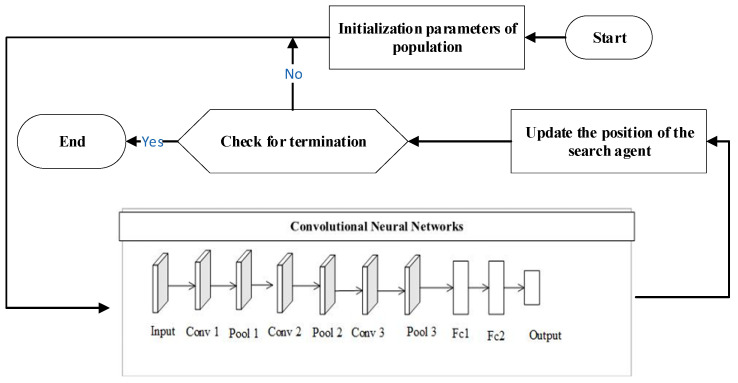
The formation process of the OLCPA-CNN model.

**Figure 4 biomimetics-08-00268-f004:**
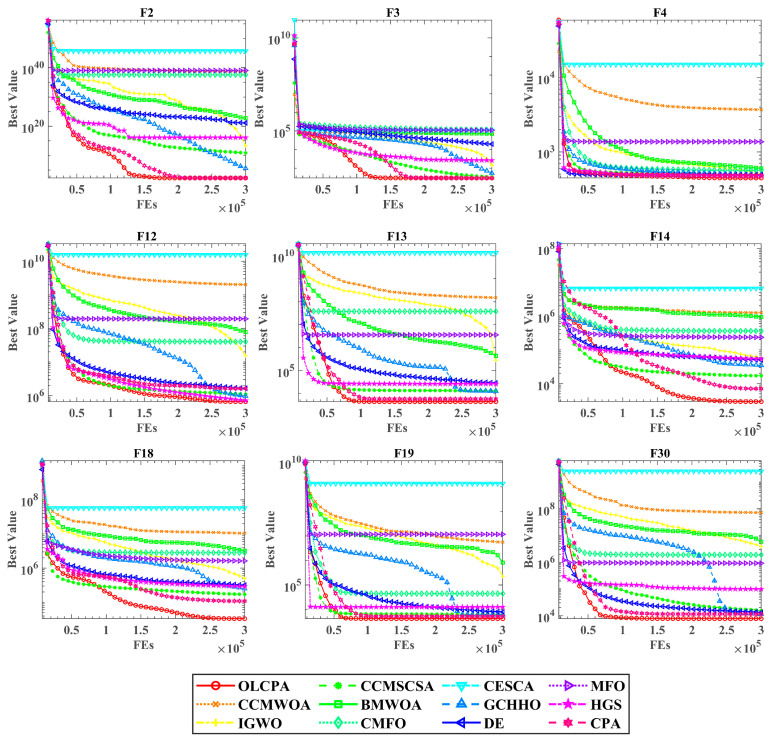
Convergence curves of OLCPA and other algorithms. In F4, OLCPA competes fiercely with HGS, but OLCPA finally finds the optimal value. In other benchmark functions, OLCPA’s convergence speed is relatively fast compared with the other algorithms. Additionally, with the increase in iterations, its exploration ability is enhanced, which means it can explore more valuable areas and finally find the optimal value.

**Figure 5 biomimetics-08-00268-f005:**
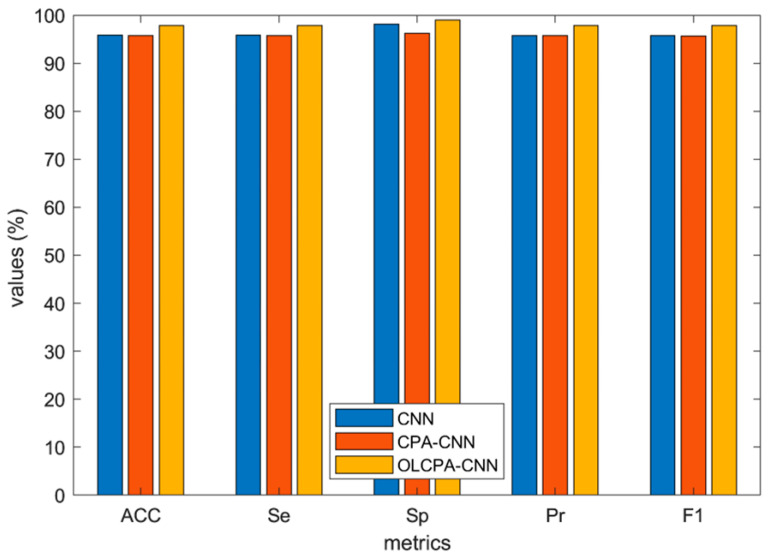
Overview of the average performance metrics of OLCPA-CNN, CPA-CNN, and the randomly generated CNN on the MIT-BIH dataset.

**Figure 6 biomimetics-08-00268-f006:**
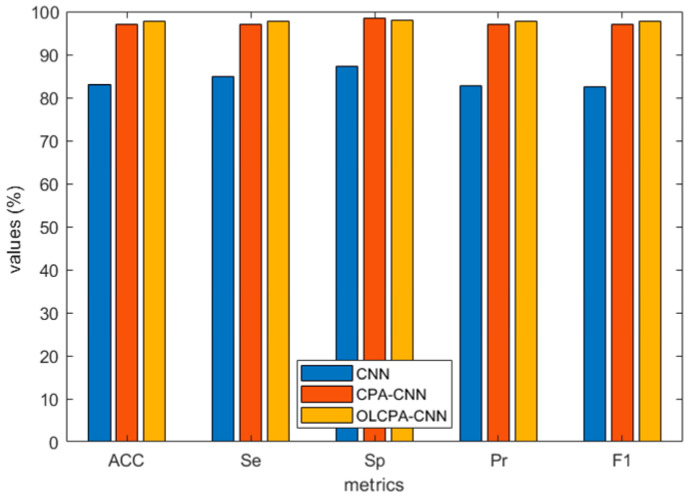
Overview of the average performance metrics of OLCPA-CNN, CPA-CNN, and the randomly generated CNN on the ST-T dataset.

**Figure 7 biomimetics-08-00268-f007:**
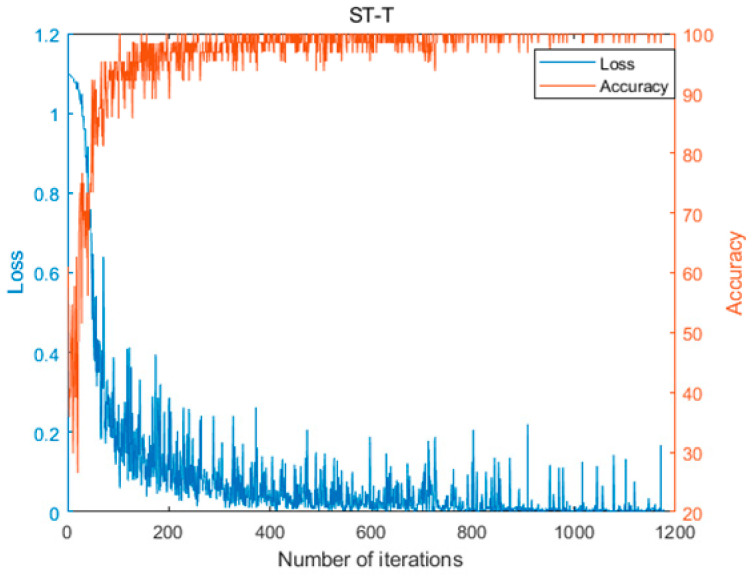
The accuracy and loss curve of OLCPA-CNN on the ST-T dataset.

**Figure 8 biomimetics-08-00268-f008:**
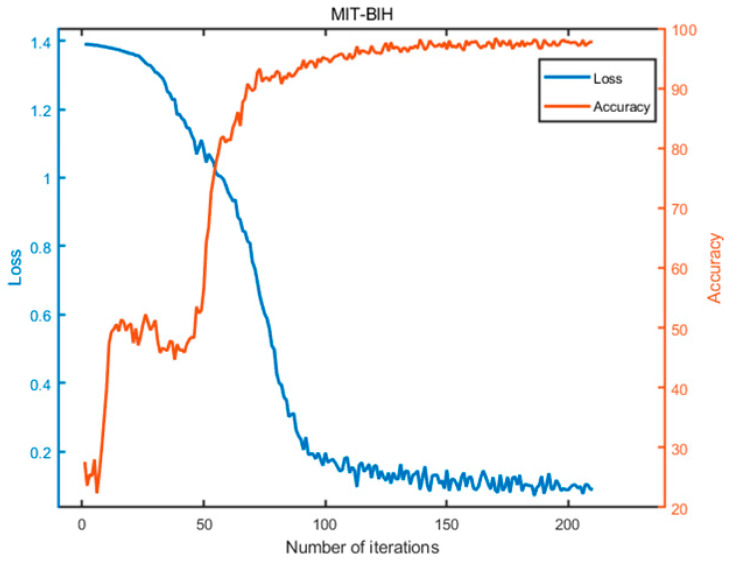
The accuracy and loss of OLCPA-CNN on the MIT-BIH dataset.

**Table 1 biomimetics-08-00268-t001:** Hyperparameter range of CNN structure.

Architecture	Hyperparameter	Range
CNN	Number of convolution kernels	[1, 15]
	Size of the convolution kernels	[1, 128]
	Number of nodes of the first fully connected layer	[0, 5000]
	Number of epochs	[1, 40]
	Learning rate	[0.0001, 0.01]
	L2 regularization	[0.001, 0.01]

**Table 2 biomimetics-08-00268-t002:** Confusion matrix.

	Actual Positive	Actual Negative
Predicted positive	*TP*	*FP*
Predicted negative	*FN*	*TN*

**Table 3 biomimetics-08-00268-t003:** Parameters of an algorithm comparison experiment.

N	D	MaxFes	Num
30	30	300,000	30

**Table 4 biomimetics-08-00268-t004:** Comparison of OLCPA and CPA dimensions.

F	Algorithm	Dim = 50		Dim = 100	
		Avg	Std	Avg	Std
F1	CPA	3.918 × 10³	5.613 × 10³	1.0134 × 10^4^	1.3351 × 10^4^
	OLCPA	2.045 × 10³	2.522 × 10³	7.4512 × 10³	6.3808 × 10³
F2	CPA	3.212 × 10³	9.123 × 10³	1.2924 × 10^13^	6.9606 × 10^13^
	OLCPA	2.804 × 10^2^	1.873 × 10^2^	2.7373 × 10^10^	1.0467 × 10^11^
F3	CPA	3.000 × 10^2^	1.723 × 10^−6^	2.3131 × 10³	1.0879 × 10³
	OLCPA	3.000 × 10^2^	1.425 × 10^−9^	3.0000 × 10^2^	2.1451 × 10^−8^
F4	CPA	5.069 × 10^2^	5.491 × 10	6.2239 × 10^2^	3.7274 × 10
	OLCPA	4.769 × 10^2^	3.367 × 10	6.3116 × 10^2^	5.2452 × 10
F5	CPA	7.157 × 10^2^	2.635 × 10	1.0656 × 10³	7.0536 × 10
	OLCPA	7.393 × 10^2^	3.785 × 10	1.1193 × 10³	5.8359 × 10
F6	CPA	6.000 × 10^2^	2.072 × 10^−4^	6.0000 × 10^2^	3.5520 × 10^−3^
	OLCPA	6.000 × 10^2^	1.946 × 10^−13^	6.0000 × 10^2^	2.6953 × 10^−13^
F7	CPA	9.926 × 10^2^	4.689 × 10	1.4700 × 10³	9.1916 × 10
	OLCPA	1.012 × 10³	4.161 × 10	1.5541 × 10³	1.2035 × 10^2^
F8	CPA	1.017 × 10³	3.768 × 10	1.3892 × 10³	7.3342 × 10
	OLCPA	1.018 × 10³	3.874 × 10	1.4321 × 10³	6.7090 × 10
F9	CPA	6.632 × 10³	1.662 × 10³	1.6982 × 10^4^	1.9060 × 10³
	OLCPA	6.566 × 10³	1.940 × 10³	1.6940 × 10^4^	1.8023 × 10³
F10	CPA	5.857 × 10³	6.643 × 10^2^	1.2755 × 10^4^	1.0274 × 10³
	OLCPA	5.598 × 10³	7.011 × 10^2^	1.2856 × 10^4^	1.2503 × 10³
F11	CPA	1.226 × 10³	3.445 × 10	1.5628 × 10³	1.0329 × 10^2^
	OLCPA	1.228 × 10³	2.865 × 10	1.5100 × 10³	1.1924 × 10^2^
F12	CPA	3.216 × 10^6^	2.379 × 10^6^	8.5478 × 10^6^	3.7663 × 10^6^
	OLCPA	2.110 × 10^6^	1.443 × 10^6^	4.4447 × 10^6^	2.1365 × 10^6^
F13	CPA	8.034 × 10³	8.404 × 10³	6.9597 × 10³	5.4080 × 10³
	OLCAP	5.223 × 10³	3.526 × 10³	4.9575 × 10³	3.4951 × 10³
F14	CPA	4.017 × 10^4^	2.045 × 10^4^	1.0750 × 10^5^	3.2124 × 10^4^
	OLCAP	8.857 × 10³	5.974 × 10³	3.4510 × 10^4^	6.5897 × 10³
F15	CPA	8.161 × 10³	5.507 × 10³	4.0322 × 10³	3.0081 × 10³
	OLCPA	9.451 × 10³	5.953 × 10³	2.7858 × 10³	1.2726 × 10³
F16	CPA	3.647 × 10³	4.486 × 10^2^	6.1075 × 10³	6.1143 × 10^2^
	OLCPA	3.453 × 10³	3.315 × 10^2^	5.9226 × 10³	5.8144 × 10^2^
F17	CPA	3.034 × 10³	3.329 × 10^2^	4.9720 × 10³	5.3734 × 10^2^
	OLCPA	3.109 × 10³	3.106 × 10^2^	4.8910 × 10³	6.0655 × 10^2^
F18	CPA	1.320 × 10^5^	2.658 × 10^4^	2.5015 × 10^5^	9.0799 × 10^4^
	OLCPA	4.246 × 10^4^	1.368 × 10^4^	1.3636 × 10^5^	2.4613 × 10^4^
F19	CPA	2.097 × 10^4^	9.306 × 10³	5.9069 × 10³	4.3932 × 10³
	OLCPA	2.660 × 10^4^	8.118 × 10³	3.8745 × 10³	1.7514 × 10³
F20	CPA	3.055 × 10³	3.027 × 10^2^	5.1363 × 10³	3.9298 × 10^2^
	OLCPA	2.891 × 10³	2.578 × 10^2^	5.1443 × 10³	4.6463 × 10^2^
F21	CPA	2.515 × 10³	3.993 × 10	2.8858 × 10³	8.8322 × 10
	OLCPA	2.527 × 10³	4.671 × 10	2.8781 × 10³	7.3269 × 10
F22	CPA	7.882 × 10³	1.718 × 10³	1.6525 × 10^4^	1.2773 × 10³
	OLCPA	7.908 × 10³	1.320 × 10³	1.6529 × 10^4^	9.8827 × 10^2^
F23	CPA	2.979 × 10³	4.502 × 10	3.1604 × 10³	6.6818 × 10
	OLCPA	3.008 × 10³	4.940 × 10	3.1599 × 10³	7.3210 × 10
F24	CPA	3.498 × 10³	1.719 × 10^2^	3.8236 × 10³	8.3982 × 10
	OLCPA	3.566 × 10³	1.472 × 10^2^	3.8711 × 10³	8.9281 × 10
F25	CPA	3.048 × 10³	4.905 × 10	3.2936 × 10³	7.0202 × 10
	OLCPA	3.052 × 10³	3.882 × 10	3.2933 × 10³	7.0629 × 10
F26	CPA	4.078 × 10³	2.100 × 10³	1.2307 × 10^4^	3.3495 × 10³
	OLCPA	5.254 × 10³	2.960 × 10³	1.3663 × 10^4^	2.7772 × 10³
F27	CPA	3.519 × 10³	1.029 × 10^2^	3.5698 × 10³	8.1696 × 10
	OLCPA	3.532 × 10³	9.428 × 10	3.6442 × 10³	8.7973 × 10
F28	CPA	3.296 × 10³	2.671 × 10	3.3820 × 10³	3.4844 × 10
	OLCPA	3.290 × 10³	2.094 × 10	3.3647 × 10³	4.4484 × 10
F29	CPA	4.233 × 10³	2.991 × 10^2^	6.7623 × 10³	4.7534 × 10^2^
	OLCPA	4.072 × 10³	2.955 × 10^2^	6.9277 × 10³	5.1706 × 10^2^
F30	CPA	9.734 × 10^5^	2.383 × 10^5^	1.4286 × 10^4^	4.3046 × 10³
	OLCPA	8.788 × 10^5^	1.518 × 10^5^	1.3530 × 10^4^	4.6178 × 10³

**Table 5 biomimetics-08-00268-t005:** Comparison of OLCPA with other algorithms.

Algorithm	F1		F2		F3	
	Avg	Std	Avg	Std	Avg	Std
OLCPA	2.6417 × 10^3^	2.6210 × 10^3^	2.0000 × 10^2^	6.1889 × 10^−6^	3.0000 × 10^2^	3.3039 × 10^−10^
CCMWOA	2.0529 × 10^10^	4.7896 × 10^9^	3.9233 × 10^38^	1.8806 × 10^39^	7.7098 × 10^4^	6.7169 × 10^3^
IGWO	1.6989 × 10^6^	8.5027 × 10^5^	2.0146 × 10^13^	8.4520 × 10^13^	1.4554 × 10^3^	6.7162 × 10^2^
CCMSCSA	3.1354 × 10^3^	3.1637 × 10^3^	7.5324 × 10^10^	2.3962 × 10^11^	3.4410 × 10^2^	3.2511 × 10^1^
BMWOA	2.0649 × 10^8^	8.6683 × 10^7^	5.7378 × 10^22^	2.6681 × 10^23^	7.0802 × 10^4^	9.5290 × 10^3^
CMFO	2.0503 × 10^8^	4.7086 × 10^8^	3.8481 × 10^37^	2.0987 × 10^38^	1.1542 × 10^5^	4.6442 × 10^4^
CESCA	5.7624 × 10^10^	4.6938 × 10^9^	5.0711 × 10^45^	1.0293 × 10^46^	1.0551 × 10^5^	1.4524 × 10^4^
GCHHO	4.6746 × 10^3^	5.9329 × 10^3^	4.0569 × 10^5^	9.1569 × 10^5^	5.5167 × 10^2^	1.6374 × 10^2^
DE	2.2872 × 10^3^	3.7753 × 10^3^	1.3602 × 10^21^	3.6101 × 10^21^	1.9826 × 10^4^	4.4953 × 10^3^
MFO	1.3517 × 10^10^	8.8111 × 10^9^	1.1274 × 10^39^	6.1676 × 10^39^	1.1049 × 10^5^	8.7936 × 10^4^
HGS	1.3861 × 10^7^	7.5867 × 10^7^	1.3521 × 10^16^	5.1457 × 10^16^	2.6774 × 10^3^	5.6921 × 10^3^
CPA	5.3939 × 10^3^	5.9328 × 10^3^	2.0098 × 10^2^	3.7267 × 10^0^	3.0000 × 10^2^	1.4672 × 10^−7^
	F4		F5		F6	
	Avg	Std	Avg	Std	Avg	Std
OLCPA	4.4457 × 10^2^	3.6133 × 10^1^	6.2701 × 10^2^	2.3849 × 10^1^	6.0000 × 10^2^	3.5452 × 10^−13^
CCMWOA	3.7003 × 10^3^	1.4526 × 10^3^	8.3497 × 10^2^	3.3283 × 10^1^	6.7164 × 10^2^	7.8115 × 10^0^
IGWO	5.0643 × 10^2^	2.3656 × 10^1^	6.1178 × 10^2^	1.6784 × 10^1^	6.2273 × 10^2^	5.5501 × 10^0^
CCMSCSA	4.9943 × 10^2^	2.7486 × 10^1^	5.8212 × 10^2^	2.3957 × 10^1^	6.0043 × 10^2^	2.9648 × 10^−1^
BMWOA	6.0019 × 10^2^	3.8438 × 10^1^	7.7892 × 10^2^	5.5275 × 10^1^	6.6611 × 10^2^	1.1204 × 10^1^
CMFO	5.6429 × 10^2^	6.6364 × 10^1^	7.2496 × 10^2^	5.0447 × 10^1^	6.5202 × 10^2^	9.3009 × 10^0^
CESCA	1.5015 × 10^4^	2.3052 × 10^3^	9.6832 × 10^2^	2.4033 × 10^1^	7.0496 × 10^2^	5.2640 × 10^0^
GCHHO	4.9548 × 10^2^	2.8019 × 10^1^	7.1025 × 10^2^	4.2271 × 10^1^	6.5178 × 10^2^	6.9046 × 10^0^
DE	4.9088 × 10^2^	9.5252 × 10^0^	6.0806 × 10^2^	9.1168 × 10^0^	6.0000 × 10^2^	0.0000 × 10^0^
MFO	1.3689 × 10^3^	8.5540 × 10^2^	7.0259 × 10^2^	6.0823 × 10^1^	6.4206 × 10^2^	1.1753 × 10^1^
HGS	4.7827 × 10^2^	2.7144 × 10^1^	6.3080 × 10^2^	2.8629 × 10^1^	6.0152 × 10^2^	1.9161 × 10^0^
CPA	4.8346 × 10^2^	2.5598 × 10^1^	6.2974 × 10^2^	2.6272 × 10^1^	6.0000 × 10^2^	1.0003 × 10^−7^
	F7		F8		F9	
	Avg	Std	Avg	Std	Avg	Std
OLCPA	8.5580 × 10^2^	3.3574 × 10^1^	9.0241 × 10^2^	1.6026 × 10^1^	2.6955 × 10^3^	5.2050 × 10^2^
CCMWOA	1.2785 × 10^3^	7.1917 × 10^1^	1.0445 × 10^3^	2.5360 × 10^1^	7.7640 × 10^3^	1.4347 × 10^3^
IGWO	9.0405 × 10^2^	5.2744 × 10^1^	8.9353 × 10^2^	2.1787 × 10^1^	2.8209 × 10^3^	8.7020 × 10^2^
CCMSCSA	8.0574 × 10^2^	1.7076 × 10^1^	9.0305 × 10^2^	2.9632 × 10^1^	9.8573 × 10^2^	7.2563 × 10^1^
BMWOA	1.1733 × 10^3^	1.0644 × 10^2^	1.0081 × 10^3^	3.0517 × 10^1^	7.2354 × 10^3^	8.8766 × 10^2^
CMFO	1.2808 × 10^3^	1.5349 × 10^2^	9.5931 × 10^2^	3.8407 × 10^1^	4.7987 × 10^3^	1.1702 × 10^3^
CESCA	1.5498 × 10^3^	5.0905 × 10^1^	1.1778 × 10^3^	1.9507 × 10^1^	1.5424 × 10^4^	1.2474 × 10^3^
GCHHO	1.0821 × 10^3^	1.0330 × 10^2^	9.4369 × 10^2^	2.0730 × 10^1^	4.7578 × 10^3^	5.8635 × 10^2^
DE	8.4129 × 10^2^	1.0450 × 10^1^	9.0777 × 10^2^	8.5623 × 10^0^	9.0000 × 10^2^	1.0765 × 10^−13^
MFO	1.1498 × 10^3^	2.0356 × 10^2^	1.0177 × 10^3^	4.6613 × 10^1^	7.1329 × 10^3^	1.7975 × 10^3^
HGS	8.9314 × 10^2^	5.1096 × 10^1^	9.0545 × 10^2^	2.2953 × 10^1^	3.5491 × 10^3^	8.3458 × 10^2^
CPA	8.4269 × 10^2^	2.7130 × 10^1^	9.0484 × 10^2^	2.1313 × 10^1^	2.3193 × 10^3^	6.1060 × 10^2^
	F10		F11		F12	
	Avg	Std	Avg	Std	Avg	Std
OLCPA	3.7605 × 10^3^	3.8850 × 10^2^	1.1756 × 10^3^	3.4123 × 10^1^	6.5992 × 10^5^	4.6976 × 10^5^
CCMWOA	7.0372 × 10^3^	6.1016 × 10^2^	3.1558 × 10^3^	5.8702 × 10^2^	2.0126 × 10^9^	1.4565 × 10^9^
IGWO	4.4687 × 10^3^	5.9061 × 10^2^	1.2642 × 10^3^	2.8641 × 10^1^	1.5414 × 10^7^	1.5084 × 10^7^
CCMSCSA	4.6758 × 10^3^	6.1466 × 10^2^	1.1870 × 10^3^	3.1743 × 10^1^	1.1060 × 10^6^	8.7007 × 10^5^
BMWOA	7.4949 × 10^3^	5.9325 × 10^2^	1.6517 × 10^3^	1.6384 × 10^2^	7.8078 × 10^7^	5.9556 × 10^7^
CMFO	7.3777 × 10^3^	1.2921 × 10^3^	4.6678 × 10^3^	3.4534 × 10^3^	4.0157 × 10^7^	1.2585 × 10^8^
CESCA	8.7430 × 10^3^	2.2735 × 10	1.0664 × 10^4^	1.6523 × 10^3^	1.5622 × 10^0^	1.5369 × 10^9^
GCHHO	5.1344 × 10^3^	6.1384 × 10^2^	1.2339 × 10^3^	5.1150 × 10^1^	9.6394 × 10^5^	7.5930 × 10^5^
DE	5.9154 × 10^3^	3.1146 × 10^2^	1.1611 × 10^3^	2.2327 × 10^1^	1.6551 × 10^6^	8.2025 × 10^5^
MFO	5.6084 × 10^3^	8.5316 × 10^2^	4.6351 × 10^3^	4.6023 × 10^3^	1.9357 × 10^8^	3.4217 × 10^8^
HGS	3.9194 × 10^3^	4.7298 × 10^2^	1.2032 × 10^3^	3.5853 × 10^1^	7.1069 × 10^5^	5.9578 × 10^5^
CPA	3.6850 × 10^3^	4.6305 × 10^2^	1.1700 × 10^3^	3.4461 × 10^1^	1.6148 × 10^6^	1.2540 × 10^6^
	F13		F14		F15	
	Avg	Std	Avg	Std	Avg	Std
OLCPA	4.4219 × 10^3^	2.9844 × 10^3^	2.9128 × 10^3^	1.0472 × 10^3^	3.2767 × 10^3^	2.4501 × 10^3^
CCMWOA	1.5038 × 10^8^	2.1857 × 10^8^	1.2576 × 10^6^	8.9707 × 10^5^	5.8975 × 10^6^	9.3662 × 10^6^
IGWO	2.8168 × 10^5^	3.9417 × 10^5^	5.3978 × 10^4^	3.4567 × 10^4^	5.6870 × 10^4^	2.9361 × 10^4^
CCMSCSA	1.3367 × 10^4^	1.1171 × 10^4^	1.6896 × 10^4^	1.5715 × 10^4^	3.0463 × 10^3^	2.0005 × 10^3^
BMWOA	4.3710 × 10^5^	7.0418 × 10^5^	9.4482 × 10^5^	8.0898 × 10^5^	1.7030 × 10^5^	2.7468 × 10^5^
CMFO	3.7917 × 10^7^	1.9343 × 10^8^	3.5943 × 10^5^	8.2977 × 10^5^	3.0292 × 10^4^	3.2825 × 10^4^
CESCA	1.3433 × 10^10^	3.9068 × 10^9^	6.5888 × 10^6^	2.8850 × 10^6^	4.5428 × 10^8^	1.8134 × 10^8^
GCHHO	1.2843 × 10^4^	1.5165 × 10^4^	3.4759 × 10^4^	2.5482 × 10^4^	6.3001 × 10^3^	6.5459 × 10^3^
DE	2.9103 × 10^4^	1.6893 × 10^4^	4.9826 × 10^4^	2.5793 × 10^4^	8.6203 × 10^3^	5.3792 × 10^3^
MFO	3.5810 × 10^6^	1.3021 × 10^7^	2.3452 × 10^5^	6.2121 × 10^5^	6.7501 × 10^4^	6.6285 × 10^4^
HGS	2.6168 × 10^4^	2.4091 × 10^4^	5.3803 × 10^4^	4.0736 × 10^4^	1.7192 × 10^4^	1.5459 × 10^4^
CPA	5.8096 × 10^3^	1.1246 × 10^4^	7.0490 × 10^3^	4.9848 × 10^3^	2.2795 × 10^3^	9.4388 × 10^2^
	F16		F17		F18	
	Avg	Std	Avg	Std	Avg	Std
OLCPA	2.5673 × 10^3^	3.5254 × 10^2^	2.0234 × 10^3^	1.5350 × 10^2^	3.3044 × 10^4^	1.4827 × 10^4^
CCMWOA	3.9526 × 10^3^	6.7820 × 10^2^	2.7756 × 10^3^	3.8592 × 10^2^	1.0532 × 10^7^	1.0505 × 10^7^
IGWO	2.5650 × 10^3^	3.5963 × 10^2^	2.0210 × 10^3^	1.4303 × 10^2^	4.9309 × 10^5^	4.1194 × 10^5^
CCMSCSA	2.5013 × 10^3^	2.6859 × 10^2^	2.0794 × 10^3^	1.8933 × 10^2^	1.6964 × 10^5^	1.5010 × 10^5^
BMWOA	3.4890 × 10^3^	5.2235 × 10^2^	2.4671 × 10^3^	2.1627 × 10^2^	3.2300 × 10^6^	3.6494 × 10^6^
CMFO	2.9376 × 10^3^	5.1792 × 10^2^	2.4441 × 10^3^	3.0889 × 10^2^	2.8221 × 10^6^	5.1454 × 10^6^
CESCA	5.9581 × 10^3^	4.8739 × 10^2^	4.4216 × 10^3^	4.3712 × 10^2^	5.7643 × 10^7^	2.6176 × 10^7^
GCHHO	2.7374 × 10^3^	2.7503 × 10^2^	2.3088 × 10^3^	2.6833 × 10^2^	2.5047 × 10^5^	3.0345 × 10^5^
DE	2.0652 × 10^3^	1.4220 × 10^2^	1.8272 × 10^3^	4.6232 × 10^1^	3.2091 × 10^5^	1.8003 × 10^5^
MFO	3.1336 × 10^3^	4.4336 × 10^2^	2.5667 × 10^3^	3.1189 × 10^2^	1.6242 × 10^6^	3.0380 × 10^6^
HGS	2.6782 × 10^3^	3.3225 × 10^2^	2.2166 × 10^3^	2.5020 × 10^2^	2.8938 × 10^5^	2.7168 × 10^5^
CPA	2.7639 × 10^3^	2.9671 × 10^2^	2.1603 × 10^3^	2.6412 × 10^2^	1.0822 × 10^5^	6.6646 × 10^4^
	F19		F20		F21	
	Avg	Std	Avg	Std	Avg	Std
OLCPA	4.2596 × 10^3^	2.0678 × 10^3^	2.3366 × 10^3^	1.1975 × 10^2^	2.4210 × 10^3^	2.6344 × 10^1^
CCMWOA	5.5422 × 10^6^	9.1972 × 10^6^	2.7663 × 10^3^	1.8365 × 10^2^	2.6152 × 10^3^	6.4209 × 10^1^
IGWO	2.2651 × 10^5^	2.5430 × 10^5^	2.3539 × 10^3^	1.2977 × 10^2^	2.3977 × 10^3^	2.4054 × 10^1^
CCMSCSA	6.5091 × 10^3^	5.1531 × 10^3^	2.3399 × 10^3^	1.2859 × 10^2^	2.3752 × 10^3^	1.8620 × 10^1^
BMWOA	8.1030 × 10^5^	1.1393 × 10^6^	2.7627 × 10^3^	1.8733 × 10^2^	2.5221 × 10^3^	5.0213 × 10^1^
CMFO	4.4672 × 10^4^	7.9129 × 10^4^	2.7796 × 10^3^	1.8148 × 10^2^	2.4958 × 10^3^	3.7613 × 10^1^
CESCA	1.3527 × 10^9^	4.4096 × 10^8^	3.1735 × 10^3^	1.3671 × 10^2^	2.7653 × 10^3^	3.4531 × 10^1^
GCHHO	6.1960 × 10^3^	5.0850 × 10^3^	2.5673 × 10^3^	1.8589 × 10^2^	2.4895 × 10^3^	5.0529 × 10^1^
DE	8.0940 × 10^3^	5.1894 × 10^3^	2.1309 × 10^3^	8.2781 × 10^1^	2.4037 × 10^3^	9.0200 × 10^0^
MFO	1.1628 × 10^7^	3.7701 × 10^7^	2.7001 × 10^3^	2.2561 × 10^2^	2.5056 × 10^3^	4.5377 × 10^1^
HGS	1.2762 × 10^4^	1.5843 × 10^4^	2.4769 × 10^3^	1.7556 × 10^2^	2.4252 × 10^3^	2.8533 × 10^1^
CPA	5.3098 × 10^3^	1.9655 × 10^3^	2.4748 × 10^3^	1.4970 × 10^2^	2.4060 × 10^3^	7.0170 × 10^1^
	F22		F23		F24	
	Avg	Std	Avg	Std	Avg	Std
OLCPA	3.9509 × 10^3^	1.9526 × 10^3^	2.7595 × 10^3^	3.2074 × 10^1^	3.1311 × 10^3^	9.8525 × 10^1^
CCMWOA	7.3798 × 10^3^	1.3564 × 10^3^	3.1950 × 10^3^	1.1166 × 10^2^	3.3448 × 10^3^	1.1703 × 10^2^
IGWO	2.3179 × 10^3^	3.7459 × 10^1^	2.7712 × 10^3^	3.0804 × 10^1^	2.9433 × 10^3^	3.3180 × 10^1^
CCMSCSA	2.3011 × 10^3^	1.8012 × 10^0^	2.7389 × 10^3^	2.2920 × 10^1^	2.9120 × 10^3^	2.9458 × 10^1^
BMWOA	6.0884 × 10^3^	3.1127 × 10^3^	2.9482 × 10^3^	7.9106 × 10^1^	3.1150 × 10^3^	7.4716 × 10^1^
CMFO	5.4333 × 10^3^	2.9116 × 10^3^	2.9734 × 10^3^	7.2165 × 10^1^	3.1313 × 10^3^	1.1677 × 10^2^
CESCA	9.5457 × 10^3^	5.7616 × 10^2^	3.4764 × 10^3^	4.8311 × 10^1^	3.4817 × 10^3^	3.3877 × 10^1^
GCHHO	4.1361 × 10^3^	2.1478 × 10^3^	2.9327 × 10^3^	6.8914 × 10^1^	3.0983 × 10^3^	7.3773 × 10^1^
DE	3.7200 × 10^3^	1.7703 × 10^3^	2.7561 × 10^3^	8.0338 × 10^0^	2.9580 × 10^3^	1.1083 × 10^1^
MFO	6.4721 × 10^3^	1.7866 × 10^3^	2.8414 × 10^3^	3.5967 × 10^1^	2.9872 × 10^3^	3.6604 × 10^1^
HGS	4.6540 × 10^3^	1.5057 × 10^3^	2.7673 × 10^3^	2.8088 × 10^1^	3.0210 × 10^3^	4.9364 × 10^1^
CPA	3.1890 × 10^3^	1.6475 × 10^3^	2.7663 × 10^3^	3.4140 × 10^1^	3.0664 × 10^3^	6.5262 × 10^1^
	F25		F26		F27	
	Avg	Std	Avg	Std	Avg	Std
OLCPA	2.8894 × 10^3^	7.7206 × 10^0^	4.4389 × 10^3^	1.1528 × 10^3^	3.2423 × 10^3^	1.9934 × 10^1^
CCMWOA	3.3774 × 10^3^	1.1165 × 10^2^	8.7361 × 10^3^	9.8681 × 10^2^	3.5982 × 10^3^	1.5901 × 10^2^
IGWO	2.9064 × 10^3^	1.6945 × 10^1^	4.8594 × 10^3^	2.8776 × 10^2^	3.2367 × 10^3^	1.3129 × 10^1^
CCMSCSA	2.9035 × 10^3^	1.6558 × 10^1^	3.5575 × 10^3^	1.1866 × 10^3^	3.2607 × 10^3^	2.5012 × 10^1^
BMWOA	3.0206 × 10^3^	3.3567 × 10^1^	6.8057 × 10^3^	1.2593 × 10^3^	3.3084 × 10^3^	5.7831 × 10^1^
CMFO	2.9541 × 10^3^	3.6301 × 10^1^	6.8104 × 10^3^	7.5311 × 10^2^	3.4320 × 10^3^	1.5391 × 10^2^
CESCA	5.5207 × 10^3^	4.9939 × 10^2^	1.1158 × 10^4^	5.9604 × 10^2^	3.6926 × 10^3^	7.6258 × 10^1^
GCHHO	2.8956 × 10^3^	1.6050 × 10^1^	6.0549 × 10^3^	1.2718 × 10^3^	3.2638 × 10^3^	2.8447 × 10^1^
DE	2.8874 × 10^3^	3.0941 × 10^−1^	4.6573 × 10^3^	7.3190 × 10^1^	3.2061 × 10^3^	3.4861 × 10^0^
MFO	3.2124 × 10^3^	3.9290 × 10^2^	5.8620 × 10^3^	4.3155 × 10^2^	3.2565 × 10^3^	2.4574 × 10^1^
HGS	2.8915 × 10^3^	1.3659 × 10^1^	4.9433 × 10^3^	3.3033 × 10^2^	3.2306 × 10^3^	1.5047 × 10^1^
CPA	2.8988 × 10^3^	1.8851 × 10^1^	4.3956 × 10^3^	1.0423 × 10^3^	3.2447 × 10^3^	2.3567 × 10^1^
	F28		F29		F30	
	Avg	Std	Avg	Std	Avg	Std
OLCPA	3.1166 × 10^3^	3.6219 × 10^1^	3.5490 × 10^3^	1.4289 × 10^2^	7.7032 × 10^3^	2.1026 × 10^3^
CCMWOA	4.5490 × 10^3^	5.1054 × 10^2^	5.3770 × 10^3^	7.8372 × 10^2^	7.2310 × 10^7^	6.3627 × 10^7^
IGWO	3.2621 × 10^3^	3.0754 × 10^1^	3.8055 × 10^3^	1.8503 × 10^2^	3.8641 × 10^6^	3.0498 × 10^6^
CCMSCSA	3.2293 × 10^3^	2.6311 × 10^1^	3.7148 × 10^3^	1.9800 × 10^2^	1.5266 × 10^4^	8.5355 × 10^3^
BMWOA	3.3944 × 10^3^	4.6263 × 10^1^	4.7907 × 10^3^	3.5255 × 10^2^	5.9215 × 10^6^	3.3072 × 10^6^
CMFO	3.3422 × 10^3^	5.8653 × 10^1^	4.5953 × 10^3^	3.6267 × 10^2^	1.8825 × 10^6^	5.1423 × 10^6^
CESCA	7.1979 × 10^3^	3.7582 × 10^2^	6.0902 × 10^3^	2.2499 × 10^2^	2.5420 × 10^9^	8.2166 × 10^8^
GCHHO	3.2262 × 10^3^	2.5508 × 10^1^	4.0266 × 10^3^	2.1016 × 10^2^	1.1463 × 10^4^	4.0577 × 10^3^
DE	3.1752 × 10^3^	4.9966 × 10^1^	3.5037 × 10^3^	6.6936 × 10^1^	1.3382 × 10^4^	3.7353 × 10^3^
MFO	4.5833 × 10^3^	9.6836 × 10^2^	4.2258 × 10^3^	2.8249 × 10^2^	9.2011 × 10^5^	1.1203 × 10^6^
HGS	3.2078 × 10^3^	5.5946 × 10^1^	3.7668 × 10^3^	1.5864 × 10^2^	9.8961 × 10^4^	1.2709 × 10^5^
CPA	3.1488 × 10^3^	5.0178 × 10^1^	3.7367 × 10^3^	1.9795 × 10^2^	1.1526 × 10^4^	4.4883 × 10^3^
	Overall rank					
	Rank	Avg	+/=/−			
OLCPA	1	3.1322	~			
CCMSCSA	2	3.6022	15/8/7			
CPA	3	3.6933	15/13/2			
DE	4	3.7878	12/9/9			
HGS	5	4.6933	19/9/2			
IGWO	6	5.4800	18/8/4			
GCHHO	7	5.7567	24/0/0			
CMFO	8	8.2333	29/1/0			
MFO	9	8.2611	29/0/1			
BMWOA	10	9.0422	29/1/0			
CCMWOA	11	1.0409	30/0/0			
CESCA	12	1.1909	30/0/0			

**Table 6 biomimetics-08-00268-t006:** The *p*-value of OLCPA and other algorithms.

	CCMWOA	IGWO	CCMSCSA	BMWOA	CMFO	CESCA
F1	1.7344 × 10^−6^	1.7344 × 10^−6^	7.1889 × 10^−1^	1.7344 × 10^−6^	1.7344 × 10^−6^	1.7344 × 10^−6^
F2	1.7344 × 10^−6^	1.7344 × 10^−6^	1.7344 × 10^−6^	1.7344 × 10^−6^	1.7344 × 10^−6^	1.7344 × 10^−6^
F3	1.7344 × 10^−6^	1.7344 × 10^−6^	1.7344 × 10^−6^	1.7344 × 10^−6^	1.7344 × 10^−6^	1.7344 × 10^−6^
F4	1.7344 × 10^−6^	4.2857 × 10^−6^	4.7292 × 10^−6^	1.7344 × 10^−6^	1.7344 × 10^−6^	1.7344 × 10^−6^
F5	1.7344 × 10^−6^	1.1748 × 10^−2^	1.1265 × 10^−5^	1.7344 × 10^−6^	1.7344 × 10^−6^	1.7344 × 10^−6^
F6	1.7344 × 10^−6^	1.7344 × 10^−6^	1.7344 × 10^−6^	1.7344 × 10^−6^	1.7344 × 10^−6^	1.7344 × 10^−6^
F7	1.7344 × 10^−6^	3.8811 × 10^−4^	2.8786 × 10^−6^	1.7344 × 10^−6^	1.7344 × 10^−6^	1.7344 × 10^−6^
F8	1.7344 × 10^−6^	3.8723 × 10^−2^	8.9364 × 10^−1^	1.7344 × 10^−6^	2.3534 × 10^−6^	1.7344 × 10^−6^
F9	1.7344 × 10^−6^	4.4052 × 10^−1^	1.7344 × 10^−6^	1.7344 × 10^−6^	2.6033 × 10^−6^	1.7344 × 10^−6^
F10	1.7344 × 10^−6^	3.4053 × 10^−5^	9.3157 × 10^−6^	1.7344 × 10^−6^	1.7344 × 10^−6^	1.7344 × 10^−6^
F11	1.7344 × 10^−6^	1.7344 × 10^−6^	1.7138 × 10^−1^	1.7344 × 10^−6^	1.7344 × 10^−6^	1.7344 × 10^−6^
F12	1.7344 × 10^−6^	1.9209 × 10^−6^	3.1603 × 10^−2^	1.7344 × 10^−6^	4.7292 × 10^−6^	1.7344 × 10^−6^
F13	1.7344 × 10^−6^	1.7344 × 10^−6^	3.3173 × 10^−4^	1.7344 × 10^−6^	6.3391 × 10^−6^	1.7344 × 10^−6^
F14	1.7344 × 10^−6^	1.9209 × 10^−6^	8.4661 × 10^−6^	1.7344 × 10^−6^	1.7344 × 10^−6^	1.7344 × 10^−6^
F15	1.7344 × 10^−6^	1.7344 × 10^−6^	9.2626 × 10^−1^	1.7344 × 10^−6^	2.1630 × 10^−5^	1.7344 × 10^−6^
F16	1.7344 × 10^−6^	5.8571 × 10^−1^	4.1653 × 10^−1^	3.5152 × 10^−6^	5.7924 × 10^−5^	1.7344 × 10^−6^
F17	1.7344 × 10^−6^	8.9364 × 10^−1^	3.8203 × 10^−1^	2.1266 × 10^−6^	9.3157 × 10^−6^	1.7344 × 10^−6^
F18	1.7344 × 10^−6^	1.7344 × 10^−6^	1.7344 × 10^−6^	1.7344 × 10^−6^	1.7344 × 10^−6^	1.7344 × 10^−6^
F19	1.7344 × 10^−6^	1.7344 × 10^−6^	3.8723 × 10^−2^	1.7344 × 10^−6^	5.2872 × 10^−4^	1.7344 × 10^−6^
F20	1.7344 × 10^−6^	6.8836 × 10^−1^	8.6121 × 10^−1^	2.6033 × 10^−6^	1.7344 × 10^−6^	1.7344 × 10^−6^
F21	1.7344 × 10^−6^	2.4147 × 10^−3^	1.2381 × 10^−5^	1.7344 × 10^−6^	2.1266 × 10^−6^	1.7344 × 10^−6^
F22	1.2381 × 10^−5^	1.0201 × 10^−1^	2.5637 × 10^−2^	4.9916 × 10^−3^	2.3038 × 10^−2^	1.7344 × 10^−6^
F23	1.7344 × 10^−6^	1.5286 × 10^−1^	2.8486 × 10^−2^	1.7344 × 10^−6^	1.7344 × 10^−6^	1.7344 × 10^−6^
F24	6.9838 × 10^−6^	2.1266 × 10^−6^	1.7344 × 10^−6^	2.9894 × 10^−1^	6.5833 × 10^−1^	1.7344 × 10^−6^
F25	1.7344 × 10^−6^	1.1499 × 10^−4^	9.7110 × 10^−5^	1.7344 × 10^−6^	1.7344 × 10^−6^	1.7344 × 10^−6^
F26	1.7344 × 10^−6^	2.2102 × 10^−1^	9.8421 × 10^−3^	1.9729 × 10^−5^	1.7344 × 10^−6^	1.7344 × 10^−6^
F27	1.7344 × 10^−6^	5.0383 × 10^−1^	9.8421 × 10^−3^	3.5152 × 10^−6^	2.6033 × 10^−6^	1.7344 × 10^−6^
F28	1.7344 × 10^−6^	1.7344 × 10^−6^	1.7344 × 10^−6^	1.7344 × 10^−6^	1.7344 × 10^−6^	1.7344 × 10^−6^
F29	1.7344 × 10^−6^	1.4936 × 10^−5^	6.6392 × 10^−4^	1.7344 × 10^−6^	1.7344 × 10^−6^	1.7344 × 10^−6^
F30	1.7344 × 10^−6^	1.7344 × 10^−6^	1.7988 × 10^−5^	1.7344 × 10^−6^	1.7344 × 10^−6^	1.7344 × 10^−6^
	GCHHO		DE	MFO	HGS	CPA
F1	4.1653 × 10^−1^		1.5886 × 10^−1^	1.7344 × 10^−6^	8.1878 × 10^−5^	4.0702 × 10^−2^
F2	1.7344 × 10^−6^		1.7344 × 10^−6^	1.7344 × 10^−6^	1.7344 × 10^−6^	1.9209 × 10^−6^
F3	1.7344 × 10^−6^		1.7344 × 10^−6^	1.7344 × 10^−6^	1.7344 × 10^−6^	1.7344 × 10^−6^
F4	3.4053 × 10^−5^		1.9729 × 10^−5^	1.7344 × 10^−6^	1.1499 × 10^−4^	4.4493 × 10^−5^
F5	3.8822 × 10^−6^		8.3071 × 10^−4^	9.3157 × 10^−6^	4.7795 × 10^−1^	4.5281 × 10^−1^
F6	1.7344 × 10^−6^		4.3205 × 10^−8^	1.7344 × 10^−6^	1.7344 × 10^−6^	1.0000 × 10^0^
F7	1.7344 × 10^−6^		7.5213 × 10^−2^	1.7344 × 10^−6^	6.6392 × 10^−4^	1.4704 × 10^−1^
F8	6.3391 × 10^−6^		1.3591 × 10^−1^	1.7344 × 10^−6^	4.0483 × 10^−1^	6.2884 × 10^−1^
F9	1.7344 × 10^−6^		1.7344 × 10^−6^	1.7344 × 10^−6^	1.6046 × 10^−4^	1.7518 × 10^−2^
F10	2.8786 × 10^−6^		1.7344 × 10^−6^	1.7344 × 10^−6^	1.7791 × 10^−1^	4.2843 × 10^−1^
F11	1.3595 × 10^−4^		8.2206 × 10^−2^	1.7344 × 10^−6^	8.2167 × 10^−3^	3.0861 × 10^−1^
F12	8.2206 × 10^−2^		3.4053 × 10^−5^	1.7344 × 10^−6^	7.1889 × 10^−1^	1.2866 × 10^−3^
F13	1.2866 × 10^−3^		1.7344 × 10^−6^	1.7344 × 10^−6^	1.9209 × 10^−6^	5.3044 × 10^−1^
F14	1.7344 × 10^−6^		1.7344 × 10^−6^	1.7344 × 10^−6^	2.1266 × 10^−6^	1.2381 × 10^−5^
F15	1.9646 × 10^−3^		8.9187 × 10^−5^	1.7344 × 10^−6^	1.2506 × 10^−4^	1.1093 × 10^−1^
F16	5.7096 × 10^−2^		1.7344 × 10^−6^	1.0570 × 10^−4^	1.2044 × 10^−1^	2.1827 × 10^−2^
F17	5.2872 × 10^−4^		1.3601 × 10^−5^	2.1266 × 10^−6^	4.6818 × 10^−3^	1.1748 × 10^−2^
F18	1.7344 × 10^−6^		1.7344 × 10^−6^	1.7344 × 10^−6^	1.9209 × 10^−6^	4.7292 × 10^−6^
F19	2.4308 × 10^−2^		1.3820 × 10^−3^	3.5152 × 10^−6^	1.1138 × 10^−3^	2.0671 × 10^−2^
F20	4.0715 × 10^−5^		1.9209 × 10^−6^	1.9209 × 10^−6^	1.0357 × 10^−3^	1.8910 × 10^−4^
F21	4.7292 × 10^−6^		2.9575 × 10^−3^	2.1266 × 10^−6^	4.4052 × 10^−1^	1.9861 × 10^−1^
F22	3.8203 × 10^−1^		9.9179 × 10^−1^	1.1499 × 10^−4^	1.5886 × 10^−1^	3.4908 × 10^−1^
F23	1.7344 × 10^−6^		5.5774 × 10^−1^	2.1266 × 10^−6^	1.4704 × 10^−1^	4.5281 × 10^−1^
F24	1.7138 × 10^−1^		1.7344 × 10^−6^	6.3391 × 10^−6^	2.5967 × 10^−5^	9.2710 × 10^−3^
F25	6.5641 × 10^−2^		6.1431 × 10^−1^	2.3534 × 10^−6^	9.5899 × 10^−1^	3.5009 × 10^−2^
F26	7.1570 × 10^−4^		9.0993 × 10^−1^	6.9838 × 10^−6^	4.9498 × 10^−2^	9.2626 × 10^−1^
F27	2.4147 × 10^−3^		1.9209 × 10^−6^	3.5009 × 10^−2^	1.7518 × 10^−2^	7.1889 × 10^−1^
F28	1.9209 × 10^−6^		3.7172 × 10^−5^	1.7344 × 10^−6^	1.6394 × 10^−5^	8.1574 × 10^−4^
F29	1.9209 × 10^−6^		1.5286 × 10^−1^	1.7344 × 10^−6^	8.1878 × 10^−5^	1.7088 × 10^−3^
F30	4.5336 × 10^−4^		3.5152 × 10^−6^	1.7344 × 10^−6^	1.7344 × 10^−6^	2.5967 × 10^−5^

**Table 7 biomimetics-08-00268-t007:** Relationship between AMMI and MIT-BIH in terms of categories.

AAMI Classes	Supraventricular Ectopic Beat (S)	Normal (N)	Ventricular Ectopic Beats (VEBs)	Unknown Beat (Q)	Fusion (F)
MIT-BIH classes	Aberrated atrial premature beat (a)	Normal beat (N)	Ventricular flutter wave (!)	Paced beat (/)	Fusion of ventricular and normal beat (F)
	Supraventricular premature beat (S)	Left bundle branch block beat (L)	Ventricular escape beat (E)	Unclassifiable beat (Q)	
	Atrial premature beat (A)	Right bundle branch block beat (R)	Premature ventricular contraction (V)		
	Nodal (junctional) premature beat (J)				
	Nodal (junctional) escape beat (j)				
	Atrial escape beat (e)				

**Table 8 biomimetics-08-00268-t008:** Number of samples per category for the datasets.

	ST-T	MIT-BIH
N	1000	2500
S	1000	2500
VEB	1000	2500
Q	-	2500

**Table 9 biomimetics-08-00268-t009:** Comparison of OLCPA-CNN with other methods on the MIT-BIH dataset.

Reference	Method	Acc	Se
Acharya et al. [54]	CNN	94.03%	96.71%
Li et al. [84]	SVM	97.30%	97.40%
Patro et al. [85]	PSO, GA, SVM, and RF	95.30%	94.00%
Proposed	OLCPA-CNN	97.90%	97.90%

## Data Availability

The numerical and experimental data used to support the findings of this study are included within the article.

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
