# Peer review of "Improved Colony Predation Algorithm Optimized Convolutional Neural Networks for Electrocardiogram Signal Classification"

_biomimetics, 2023, doi:10.3390/biomimetics8030268_

Round 1

Reviewer 1 Report

In this paper, the OLCPA-CNN, an improved Colony Predation Algorithm (CPA), was proposed and applied to the parameter optimization of Convolutional Neural Network (CNN). The OLCPA-CNN was compared with other state-of-the-art methods on different datasets, and the results showed its effectiveness outperformed other algorithms. This paper is rich in content and the experiments are comprehensive. However, there are still some improvements to this manuscript.

1. There are some grammar and spelling errors in the article, and I suggest that the author carefully revise them. For example:

1) The sentence " Deep learning efficiently extracts features better than traditional machine learning" in line 32 is should be modified to “Deep learning is known to efficiently extract features compared to traditional machine learning approaches”

2) "GooleLeNet" in line 42 should be "GoogleLeNet"

3) The word 'these' in the sentence 'these motivates us to propose an improved CPA' on line 66 should be modified to 'this'.

4) "ST-t" in line 366 should be corrected to "ST-T".

2. There are some formatting issues in the article, and I suggest that the author revise them. For example:

1) the architecture of 9-layer CNN" in the caption of Figure 1 has inconsistent font formatting for the word "the" and the rest of the text. I suggest the author to make "the" consistent with the rest of the text by using the same font format for both in the caption of Figure 1.

2) The arrow formatting and font styles in Figures 2 and 3 are inconsistent. I suggest the author to make the arrow styles and font formatting consistent in both Figure 2 and Figure 3

3) Table 9 has a different format than the other tables in the article. I suggest the author to modify the formatting of Table 9 to match the format of Table 1.

3. In this article, variables in equations should be written in italics. For example, "w" in line 173 should be changed to "w". I suggest the author to review the entire document and make similar corrections. 

Please check the grammar.

Author Response

The comments of review #1

Comment 1:

There are some grammar and spelling errors in the article, and I suggest that the author carefully revise them. For example:

1) The sentence " Deep learning efficiently extracts features better than traditional machine learning" in line 32 is should be modified to “Deep learning is known to efficiently extract features compared to traditional machine learning approaches”

2) "GooleLeNet" in line 42 should be "GoogleLeNet"

3) The word 'these' in the sentence 'these motivates us to propose an improved CPA' on line 66 should be modified to 'this'.

4) "ST-t" in line 366 should be corrected to "ST-T".

Thank you for your valuable comment and kind suggestion. We have made the necessary revisions to the issue mentioned above, as well as addressed similar issues throughout the entire article.

Comment 2:

There are some formatting issues in the article, and I suggest that the author revise them. For example:

1) “the architecture of 9-layer CNN" in the caption of Figure 1 has inconsistent font formatting for the word "the" and the rest of the text. I suggest the author to make "the" consistent with the rest of the text by using the same font format for both in the caption of Figure 1.

2) The arrow formatting and font styles in Figures 2 and 3 are inconsistent. I suggest the author to make the arrow styles and font formatting consistent in both Figure 2 and Figure 3

3) Table 9 has a different format than the other tables in the article. I suggest the author to modify the formatting of Table 9 to match the format of Table 1.

Thank you for your valuable comment and kind suggestion. We have made the necessary revisions to the issue mentioned above, as well as addressed similar issues throughout the entire article.

Comment 3:

In this article, variables in equations should be written in italics. For example, "w" in line 173 should be changed to "w". I suggest the author to review the entire document and make similar corrections.

Thank you for your valuable comment and kind suggestion. The variable "w" has been changed to "w", and we have reviewed the entire article to address similar issues with formula variables.

Author Response

The comments of review #2

Comment 1:

Abstract contains too many non-expanded abbreviations. It should be written in a way that requires the least possible use of acronyms. Please rewrite it in a clearer way.

Thank you for your valuable comment and kind suggestion, and you have raised an important point. We have removed unnecessary abbreviations and retained necessary ones in the abstract. However, we cannot expand proprietary terms such as MIT-BIH and European ST-T.

Comment 2:

There must be major qualitative and quantitative results in the abstract. For the time being abstract lacks quantitative results.

Thank you for your valuable comment and kind suggestion, and you have raised an important point. We have added qualitative and quantitative descriptions of the results in the abstract.

Comment 3:

Line 65: Therefore, these motivates us to propose an improved CPA, and use it to optimize the parameters of the CNN

Thank you for your valuable comment and kind suggestion, and you have raised an important point. We have modified the sentence "Therefore, these motivates us to propose an improved CPA, and use it to optimize the parameters of the CNN" by changing "these" to "this".

Comment 4:

Explain IEEE CEC 2017 benchmark functions in the introduction.

Thank you for your valuable comment and kind suggestion, and you have raised an important point. We have already introduced the IEEE CEC2017 benchmark functions in the introduction.

Comment 5:

I suggest merging sections 1 and 2. Authors started with literature review already in the beginning of section 1 and then again there is a special section considered for this!

Thank you for your valuable comment and kind suggestion, and you have raised an important point. We have merged Sections 1 and 2.

Comment 6:

Literature review should not only repeat the title of published papers!!! It must be meaningful and pretty relevant to the topic of current research. Thus, I suggest digging more into cited papers in literature review and extracting the main message with a brief on the results both quantitative and qualitative. Saying the results were good, better, excellent is very general. Yet, we are writing a specialized manuscript for experts!

Thank you for your valuable comment and kind suggestion, and you have raised an important point.

We have qualitatively and quantitatively described the cited references.

Comment 7:

Line 105: To reduce this costly waste, many researchers use optimization algorithms to implement automatic 106 searches for optimal parameters of deep learning network structures.

Thank you for your valuable comment and kind suggestion, and you have raised an important point. As we have merged Sections 1 and 2 based on previous suggestions, this sentence has already been removed from the article.

Comment 8:

Line 178: When the prey begins to chase prey there are two strategies.

Thank you for your valuable comment and kind suggestion, and you have raised an important point. We have modified the sentence "When the prey begins to chase prey there are two strategies" by changing "prey" to "predator".

Comment 9:

Caption in Figure 4 should be self-explanatory. Explain the main message of figure in caption. Increase a lot the font size of titles, axes etc. in this figure to be more visible.

Thank you for your valuable comment and kind suggestion, and you have raised an important point. We have added explanatory information about the image in the caption of Figure 4.

Comment 10:

Keywords should be written in alphabetical order.

Thank you for your valuable comment and kind suggestion, and you have raised an important point. We have written the keywords in alphabetical order.

Comment 11:

The linguistic quality needs improvement. It is essential to make sure that the manuscript reads smoothly- this definitely helps the reader fully appreciate your research findings. There are grammar and writing style errors that should be corrected by the authors.

Thank you for your valuable comment and kind suggestion, and you have raised an important point. We have reviewed and corrected the grammar and spelling mistakes in the manuscript.

Comment 12:

Equations should be used with correct citations. They seem as if they are proposed and used firstly in this paper.

Thank you for your valuable comment and kind suggestion, and you have raised an important point. The equations presented in the paper are not new and have been used previously in relevant literature. We have provided appropriate citations for these sources in the text.

Comment 13:

Explain the difference between datasheets: MIT-BIH dataset and ST-T dataset. How do they differ from each other leading to different values?

To ensure consistency in the categories of heart rate types within the MIT-BIH and ST-T databases, we utilized the AAMI[1] standard to reclassify the data. However, due to differences in sample sizes between the standardized datasets, OLCPA-CNN produced varying results when tested on the two databases.

(Stergiou, G.S., et al., A Universal Standard for the Validation of Blood Pressure Measuring Devices: Association for the Advancement of Medical Instrumentation/European Society of Hypertension/International Organization for Standardization (AAMI/ESH/ISO) Collaboration Statement. Hypertension, 2018. 71(3): p. 368-374.)

Comment 14:

Some more recommendations and conclusions should be discussed about the paper considering the experimental results. The Conclusion section is weak.

Thank you for your valuable comment and kind suggestion, and you have raised an important point. We have added some conclusions and suggestions about this study in the conclusion section.

Comment 15:

Some paragraphs are too long to read. They should be divided into two or more for comprehensibility and readability.

Thank you for your valuable comment and kind suggestion, and you have raised an important point. We have divided longer paragraphs in the entire document into two or more separate paragraphs.

Comment 16

How the constraints (intervals of the attributes - lower and upper limits violation, constraint functions for the focused problem) are coped with the proposed optimization method is not clear. Is this constraint handling same for all compared methods?

In our study, we used the population dimension to control the hyperparameters of CNN. The number of dimensions indicates the number of hyperparameters, while their upper and lower limits define the corresponding hyperparameter ranges. We discussed this approach in Section 4.2. Our methodology was similar to the latter two methods compared, both of which utilized optimization algorithms to optimize network architectures for application to the MIT-BIH dataset. However, the first method compared involved manually setting the CNN structure before applying it to the MIT-BIH dataset.

Comment 17:

How do authors consider the parameters of optimized methods?

Two main approaches for optimizing CNN are structural optimization and hyperparameter optimization. While structural optimization can be a complex process, hyperparameter optimization is typically more straightforward. In this study, we referred to existing literature and focused on optimizing key parameters known to have a significant impact on CNN performance.

Comment 18:

What are the other possible methodologies that can be used to achieve your objective in relation in this work?

Besides the research presented in this paper, there are other methods that can achieve the research objectives, such as using grid search algorithm to optimize the hyperparameters of CNN or manually designing CNN models. However, both of these methods are non-automatic and require a high computational cost.

Comment 19:

Discuss your position on the generalizability of your results.

Thank you for your valuable comment and kind suggestion, and you have raised an important point.

We have discussed our position regarding the generalizability of the results of this study in the conclusion section.

Comment 20:

Clarifying the study’s limitations allows the readers to better understand under which conditions the results should be interpreted. A clear description of limitations of a study also shows that the researcher has a holistic understanding of his/her study. However, the authors fail to demonstrate this in their paper. The authors should clarify the pros and cons of the methods. What are the limitation(s) methodology(ies) adopted in this work? Please indicate practical advantages, and discuss research limitations. The discussion of the results needs to include the strengths and

weaknesses of the proposed algorithm. These limitations can be organized around simple distinctions of the choices you made in your study regarding who, what, where, when, why, and how. To have an unbiased view in the paper, there should be some discussions on the limitations of the proposed method.

Thank you for your valuable comment and kind suggestion, and you have raised an important point. We have discussed the limitations of this paper in the conclusion section.

Comment 21:

Discussion is absent. It would be better first that authors highlight their findings in the form of statements along with the conclusive data of statistical importance; mention how their findings are unique and novel; how these findings are in consensus with the existing values/ reports or how different are they from the already reported findings. Discuss your findings in terms of what was previous known and not know about the focus of your research. Did your findings cohere and/or contrast with previous research on similar groups, locations, people, etc.? Findings should respond to the purpose of the study and should be presented systematically.

Thank you for your valuable comment and kind suggestion, and you have raised an important point.

We have added a discussion section in the seventh part of the article.

Comment 22:

 I recommend the authors to review other recently developed works.

Thank you for your valuable comment and kind suggestion, and you have raised an important point. We referenced recent publications that are relevant to our work in the introduction.

Comment 23:

All figures should follow the same font and notation; the authors need to make sure all images are following the format.

Thank you for your valuable comment and kind suggestion. We have standardized the font and formatting of all images.

Comment 24:

Discuss the associated errors to your data such as MAE, RMSE and specially MRE.

Thank you for your valuable comment and kind suggestion. This article focuses on the OLCPA-CNN model for solving electrocardiogram classification problems rather than regression problems, and therefore there is no need to discuss MAE, RMSE, and MRE.

Round 2

Reviewer 2 Report

The authors have addressed almost all of the comments in a convincing way. Therefore, the manuscript is now suitable for publication in Biomimetics with minor revisions as below:

1- the quality of Figure 4 must be increased. Increase font size. As of now, this figure is not suitable for publication.

2- Although your problem is classification, since you use numbers and accuracies, therefore you should still discuss the associated errors to your data such as MAE, RMSE, and especially MRE.

Author Response

Comment 1:

The quality of Figure 4 must be increased. Increase font size. As of now, this figure is not suitable for publication.

Thank you for your valuable comment and kind suggestion, and you have raised an important point. We have re-uploaded the high-quality Figure 4, and the font size of the axes in Figure 4 has been increased.

Comment 2:

Although your problem is classification, since you use numbers and accuracies, therefore you should still discuss the associated errors to your data such as MAE, RMSE, and especially MRE.

Thank you for your valuable comment and kind suggestion, and you have raised an important point.

Sections 5.2 and 5.3 of the paper focus on addressing algorithm improvement, which is a problem of continuity. Therefore, indicators such as mean (Avg) and standard deviation (Std) are used to evaluate the performance of the OLCPA algorithm. However, in Sections 6.2 and 6.3, the practical application of the OLCPA algorithm is discussed, specifically, using OLCPA to optimize CNN hyperparameters and classifying ECG electrocardiogram signals using the OLCPA-CNN model, which is a classification problem. Hence, the article employs common metrics in categorical problems such as accuracy (ACC), precision (Pr), sensitivity (Sp), specificity (Se), and F1-score to evaluate the classification performance of the model. We have already added relevant explanations in Section 6.2.